# Specific Circular RNA Signature of Endothelial Cells: Potential Implications in Vascular Pathophysiology

**DOI:** 10.3390/ijms25010680

**Published:** 2024-01-04

**Authors:** Leïla Halidou Diallo, Jérôme Mariette, Nathalie Laugero, Christian Touriol, Florent Morfoisse, Anne-Catherine Prats, Barbara Garmy-Susini, Eric Lacazette

**Affiliations:** 1U1297-I2MC, INSERM, University of Toulouse, 1 Avenue Jean Poulhes, BP 84225, 31432 Toulouse, France; lhddiallo@yahoo.fr (L.H.D.); nathalie.laugero@inserm.fr (N.L.); florent.morfoisse@inserm.fr (F.M.); anne-catherine.prats@inserm.fr (A.-C.P.); barbara.garmy-susini@inserm.fr (B.G.-S.); 2MIAT, University of Toulouse, INRAE, 31326 Castanet-Tolosan, France; jerome.mariette@inrae.fr; 3UMR1037 INSERM, University of Toulouse, 2 Avenue Hubert Curien, 31100 Toulouse, France; christian.touriol@inserm.fr

**Keywords:** circRNA, endothelial cell, (lymph)angiogenesis, gene signature

## Abstract

Circular RNAs (circRNAs) are a recently characterized family of gene transcripts forming a covalently closed loop of single-stranded RNA. The extent of their potential for fine-tuning gene expression is still being discovered. Several studies have implicated certain circular RNAs in pathophysiological processes within vascular endothelial cells and cancer cells independently. However, to date, no comparative study of circular RNA expression in different types of endothelial cells has been performed and analysed through the lens of their central role in vascular physiology and pathology. In this work, we analysed publicly available and original RNA sequencing datasets from arterial, veinous, and lymphatic endothelial cells to identify common and distinct circRNA expression profiles. We identified 4713 distinct circRNAs in the compared endothelial cell types, 95% of which originated from exons. Interestingly, the results show that the expression profile of circular RNAs is much more specific to each cell type than linear RNAs, and therefore appears to be more suitable for distinguishing between them. As a result, we have discovered a specific circRNA signature for each given endothelial cell type. Furthermore, we identified a specific endothelial cell circRNA signature that is composed four circRNAs: circCARD6, circPLXNA2, circCASC15 and circEPHB4. These circular RNAs are produced by genes that are related to endothelial cell migration pathways and cancer progression. More detailed studies of their functions could lead to a better understanding of the mechanisms involved in physiological and pathological (lymph)angiogenesis and might open new ways to tackle tumour spread through the vascular system.

## 1. Introduction

All organs are interconnected by a complex network formed by the vascular system. It is made up of vessels that carry blood and lymph throughout the body. The blood system is a bidirectional circuit where blood is carried by arteries to deliver oxygen and nutrients to the organs’ tissue and circles back through veins for recycling and interorgan communication [1,2]. The lymphatic system has a unidirectional flow where the lymphatic fluid is carried out from tissues to the veinous circulation by the lymphatic vessels. It drains excess fluids and macromolecules from the interstitial compartment, regulating tissue homeostasis, immune surveillance, and dietary fat transport [3]. Thus, the vascular system dynamically maintains tissue homeostasis in response to physiological or pathological changes. At the interface of this system are the endothelial cells (ECs) that cover the inner wall of the vessels. ECs control the flow of fluid, cells, and substances into and out of tissues [4]. These cells also play a very prominent physiological role as they are implicated in coagulation, fibrinolysis, the regulation of vascular tone, immune response, nutrient exchange, and organ development. ECs are also key participants in multiple physiological developmental processes such as the initial formation of blood and lymphatic vessels, as well as their remodelling and maintenance later in adult life by (lymph)angiogenesis [5,6]. As such, ECs are not seen as a passive barrier, but rather as an active multifunctional tissue. Under pathophysiological conditions, ECs can be subjected to various stresses, like inflammation or hypoxia. Many pathological conditions are associated with endothelial cell dysfunction and/or alter their behaviour, including atherosclerosis [7,8], coronary artery diseases, lymphedema [9,10], inflammation [11,12], infectious and immune diseases [13], fibrosis, as well as solid tumour metastasis [14,15]. Hence, ECs are essential elements for the understanding of these pathophysiological processes.

Therefore, there is a need to better understand the characteristics of these cells to identify their biological properties at the molecular level. First, ECs are divided into blood endothelial cells (BECs) and lymphatic endothelial cells (LECs) with common, for example CD31 (PECAM-1), and specific molecular markers. Like CD31, some of the most known markers are also expressed in non-EC cell types such as platelets or macrophages, making EC isolation challenging. Furthermore, it is well documented that depending on the tissue and organ, endothelial cells can vary from each other [16]. Studies have been published which clearly demonstrate that the vascular system is locally specialized to adapt to its environmental conditions in the different tissues of various organs, and in pathological conditions like tumorigenesis [17,18]. In these studies, gene profiling has revealed the diversity of endothelial cells and specific gene expression profiles. More recently, single-cell transcriptomic studies have unveiled, on one hand, the existence of organ-specific EC molecular signatures within the same individual [19], and, on the other hand, a sex- and age-associated gene expression profile [20]. Although significant efforts have been dedicated to identifying coding and some noncoding gene marker signatures for endothelial cells [21,22], to date, no study has ever provided a systematic characterization of the newest class of RNAs called circular RNAs and their signatures in endothelial cells. Recent investigations have primarily focused on exploring the diversity of circular RNAs within hematopoietic cells [23] or in a single type of EC [24].

Circular RNAs (circRNAs) are a special class of transcripts only recently acknowledged by the scientific community in eukaryotic cells. Unlike all other classes of RNAs, circRNAs form covalently closed circular molecules, hence lacking a 5′ or 3′ end. They are generated by an alternative splicing mechanism of premature messenger (pre-mRNAs) or long noncoding RNAs (lncRNAs) called backsplicing. Backsplicing joins a 5′ splicing donor site with an upstream 3′ splicing acceptor site, in the reversed order of canonical splicing [25]. Therefore, circRNAs exhibit a significant sequence overlap with the mRNAs or lncRNAs from their parental genes, retaining specific functional sites and tissue expression patterns. However, backsplicing creates truncated and reordered sequences with potential functional consequences. This alternative splicing mechanism was originally documented three decades ago [26,27,28], but was often imputed to mis-splicing events or dismissed as creating inert splicing by-products. Twenty years later, the development of high-throughput sequencing enabled the more accurate detection, quantification, and characterization of circular RNAs [29,30,31]. CircRNAs predominantly originate from coding genes in eukaryotic cells, with the majority consisting of exonic sequences. The circular conformation gives them distinct biochemical characteristics, among which are a resistance to degradation by exoribonucleases like RNase R [32]. As a result, circular RNAs have an increased half-life compared to their linear counterparts. Functionally, these circular transcripts can directly regulate gene expression at the transcriptional level in the nucleus [33] or at the post-transcriptional level in the cytoplasm by acting as a sponge for microRNAs (miRNAs) and RNA-binding proteins (RBPs) [34,35,36]. They can also encode functional peptides because of their shared sequences with mRNA and cap-independent translation initiation mechanisms [37,38,39].

In ECs, multiple circRNAs have been identified in HUVECs [40] and in HUAECs [24]. To date, only three circRNAs expressed in endothelial cells have been studied for functional characterisation. Interestingly, it has been shown that the expression of circRNAs in HUVECs is a dynamic process as it is regulated by hypoxic stress [40]. First, Boeckel and collaborators showed that the expression of circRNAs in HUVECs is a dynamic process as it is regulated by hypoxic stress [40]. They identified the hypoxia-induced circRNA cZNF292, which was found to reduce the angiogenic response in an in vitro model and was more recently shown to participate in the morphological adaptation of aortic endothelial cells to laminar flow in vivo [41]. Second, Shan and collaborators studied circHIPK3, whose expression aggravates the vascular dysfunctions associated with diabetes in the retina [42]. Lastly, Wu and collaborators investigated the vascular endothelial cell-enriched circRNA circGNAQ. They discovered that circGNAQ protects ECs from senescence and the progression of atherosclerotic lesions by acting as a miR-146a-5p sponge [43]. None of these circular transcripts are exclusive to ECs; circHIPK3 is even among the top most abundant circular RNAs across a vast number of cell types [29].

Circular RNAs have been extensively explored in tumour cells or tissues [44,45], and many studies are currently investigating the influence of circRNAs derived from tumour cells on the vascular microenvironment. For instance, the elevated expression of circEHBP1 in bladder cancer shows a positive correlation with lymphatic metastasis and an unfavourable prognoses in patients [46]. CircEHBP1 induces the overexpression of the TGF beta receptor (TGFbR1) by physically binding to miR-130a-3p, thereby antagonising its suppressor effect. TGFbR1 overexpression mediated by circEHBP1 activates the TGF-b/SMAD3 signalling pathway, promoting VEGF-D secretion; this leads to tumour lymphangiogenesis and the lymphatic spreading of bladder cancer cells. Despite the crucial role of tumour vascularization in cancer development, progression, and metastasis, the evaluation of circRNA expressions in endothelial cells themselves from the perspective of cancer biology is still pending.

In this study, we explored RNA sequencing data from different endothelial cells to assess their circRNA expression landscape, to account for their diversity, and to identify potential new gene signatures and EC-specific markers. Our analysis reveals that a small number of circRNAs are commonly exclusive to ECs in the study and demonstrates that ECs do have a very specific circRNAs expression profile.

## 2. Results

### 2.1. EC circRNA De Novo Identification

We searched public databases for available sequencing data for primary cultures of human endothelial cells. To proceed with the identification of circRNAs in these cells, we selected total RNA paired-end sequencing datasets with sufficient depth and biological replicates to allow the identification of circRNAs through the backspliced junctions and statistical analyses. This allowed us to select datasets for HUVEC (Human Umbilical Vein Endothelial Cells), HUAEC (Human Artery Endothelial Cells), HCAEC (Human Coronary Artery endothelial Cells) and HDLEC (Human Dermal Lymphatic Endothelial Cells). Non-endothelial cell controls included Smooth Muscle Cells from the Umbilical Artery (SMC-UA) and the Stromal Vascular Fraction from Superficial Adipose Tissue (SAT-SVF) (see material and methods). The de novo identification of circRNAs followed a standard protocol, using the CIRI2 algorithm [47] (refer to the material and methods section and Appendix A). This algorithm quantifies circRNAs by detecting and counting backspliced junctions, which are non-colinear and typically found within reads that do not align to the genome. CIRI2 retrieves this “discarded” information to identify circular transcripts. The results of the analysis are shown in Figure 1A. Therefore, a total of 4713 distinct circRNAs were identified across all cell types. Among them, 2254 circRNAs (47.82%) were exclusive to a particular cell type, demonstrating the cell-specific circRNA expression profiles. We note that the number of detected circRNAs significantly varies between the cell types, for both ECs and non-Ecs: 207 circRNAs for HCAECs, 259 for SAT-SVFs, 483 for HUVECs, 706 for HDLECs, 1570 for SMC-UAs, and 2685 for HUAECs (Figure 1A). These differences can be explained in part by the variation in the sequencing depth. For this reason, we will compare the normalised read counts for each transcript between cell types in the next sections, to abrogate this bias.

The global analysis of the identified circRNAs’ genomic origin displays a typical profile, as described in the literature so far [30,31]. The overwhelming majority are exonic circRNAs (95.04%) containing about 1 to 10 exons, while intronic and intergenic circRNAs represent 3.57% and 1.39%, respectively (Figure 1B). We find a very similar distribution within each cell type (Appendix A). When we look at the chromosomal distribution, we notice a seeming correlation between the number of circular transcripts produced by the chromosome and the size of the chromosomes (Figure 1C). Chromosome 1 produces the highest number of circRNAs, with 246 for HUAECs, 153 for SMC-UAs, 58 for HDLECs, 40 for HUVECs, 33 for SAT-SVFs, and 24 for HCAECs, roughly following the decrease in sequencing depth. The two sexual chromosomes express very few circular RNAs in our samples, except for the X chromosome in the two umbilical artery-derived cells (HUAECs and SMC-UAs). This can be linked to their site of origin. There are almost no Y chromosome-derived circRNAs, even though the cells donors were both male and female. When we shift the focus to the number of circRNAs expressed per gene, we find that 73.7% of the circRNA-expressing genes only produce one circular isoform, 17.03% produce two circRNA isoforms, 4.77% produce three circRNAs, 2.26% produce four circRNAs, and 2.23% produce five to fifteen circular RNA isoforms (Figure 1D). Again, this profile is consistent with the literature, as most circRNAs arise from the circularisation of a single exon flanked by long intronic sequences rich in repeated sequences [30,48]. These results show that the RNA sequencing datasets analysed here have a similar general profile compared to the works found in the literature, which allows us to serenely move forward with the study.

### 2.2. Expression Profiles of circRNAs Prove More Adept at Discerning Distinctions between Endothelial Cells (ECs) Than Their Linear RNA Counterparts

To determine whether circular RNAs have a specific expression pattern compared to other transcripts, we analysed the expression profile of the linear RNAs in our datasets and divided them into protein-coding mRNAs and long noncoding RNAs (Figure 1A and Figure 2A). The differences in the sequencing depth do not fully explain this variability, as shown by the normalised number of circRNAs detected per million reads: 22.62 for HCAECs, 26.16 for SAT-SVFs, 11.38 for HUVECs, 6.85 for HDLECs, 10.3 for SMC-UAs, and 18.59 for HUAECs (Figure 2A). A more meaningful comparison involves evaluating, for each cell type, the normalized number of circular RNAs against the total transcripts detected (Figure 2A). This analysis reveals that among ECs, HUAECs express the highest number of circRNAs, followed by HDLECs, HUVECs and HCAECs. Circular RNAs represent, respectively, 19.48%, 6.05%, 4.92%, and 2.96% of the total transcripts detected in those endothelial cells. Among non-ECs, SMC-UAs generate circular transcripts at a rate of 13.27%, whereas SAT-SVFs exhibit a lower production of only 3.79%. Knowing that lymphatic endothelial cells have a veinous origin [49], and setting aside the case of HCAECs affected by a low sequencing depth, it seems that arterial cell types produce more circular transcripts than veinous cell types at comparable sequencing depths. Interestingly, the number of expressed circRNAs is much more variable across cell types than the number of lncRNAs and mRNAs: a mean of 16 expressed circRNAs per million reads with a standard deviation (SD) of 8 across cell lines, a mean of 128 expressed lncRNAs with a SD of 125, and a mean of 185 mRNAs with a SD of 183 (Appendix A). When we compare the coefficient of variance of circRNA expression with that of lncRNAs and mRNAs between the different cell types, there is a significant difference (*p* < 0.0001, Figure 2B). The same is true when comparing circRNA with the matched linear RNA from the same gene between all cell types (*p* < 0.05). If we compare the circRNA, mRNA and lncRNA expression variance between endothelial cells only, the differences are also significant (*p* < 0.0001). These results suggest that the variation in the circular RNA expression levels between cell types, specifically ECs, is at least partially independent of the variation in the associated linear transcripts levels. This hints towards the existence of endothelial-cell-type-specific circRNA expression patterns. We generated a density plot overlapping the amount of circRNAs with a high expression variance (>75%) alongside their corresponding linear counterparts, which also exhibit high expression variance (Figure 2C). Notably, there are regions in which the two curves diverge, reinforcing the observation of distinct circRNA expression profiles specific to endothelial cells.

Studies on circular transcripts commonly use the ratio of circRNA backspliced/junction reads on matched linear reads to assess the relative abundance of each circRNA compared to the main linear transcript produced by the same gene. This value is called the CLR for the circular/linear ratio and is one of the CIRI2 algorithm outputs. The comparison of the CLR between cell types shows that all differences are significant (*p* < 0.05 Figure 2D and Appendix A). Within ECs, HCAEC circRNAs are the most abundant and represent 62% of their linear counterpart on average. For these, the low sequencing depth results in the detection of the most abundant circRNAs, which tilts the CLR towards higher values, since less abundant circRNAs are not detected at all. For HUVECs, HUAECs and HDLECs, circRNAs represent 31%, 20% and 14% of the matched linear RNA level on average. Although HUAECs express the highest number of distinct circRNA isoforms, in HUVEC, the few expressed circRNAs are more abundant than in other ECs. There is no significant correlation between the number of circRNA reads and the number of matched linear transcripts reads (Appendix A). This shows that the number of circular transcript molecules produced by a gene is mainly independent of the simple transcription rate of the locus in those cells. CircRNAs are not just a by-product of the splicing mechanisms in endothelial cells, their identity and abundance are a specific feature of each cell type.

The heatmap representation of all RNA sequencing data, along with the Pearson correlation matrix (Figure 3), provides additional evidence supporting these findings. Notable correlation patterns emerge when examining circRNAs, their linear counterparts, lncRNAs, or mRNA expression across different cell types (Figure 3B and Appendix A). The mRNA expression profiles of ECs are largely more correlated with each other (55% to 84%) than with non-ECs (32% to 58%, except for HCAECs), which testifies to the relevance of the data sets collected. For circular RNAs, HDLECs, HUVECs and HUAECs have the highest correlation of expression profiles (60% to 77%), with the two veinous-associated cell types being the closest. In contrast, the linear RNAs from the same genes have a very low correlation rate (10% to 23%), which supports the hypothesis that circRNA biogenesis is regulated by cell-type-specific mechanisms in ECs. The correlation matrix of circular RNAs displays a sharper contrast between ECs. For instance, HDLECs have a 77% circRNA expression correlation with HUVECs and 60% with HUAECs, while this figure, respectively, is 23% and 16% for the associated linear RNAs, 91% and 96% for the lncRNAs, and lastly 78% and 73% for mRNAs. Circular RNA levels are thus much better at distinguishing between two cell types, compared to linear RNAs. The only exception pertains to circRNAs in HCAECs, likely influenced by the sequencing depth, an aspect to consider during the validation phase.

All in all, these results confirm the existence of EC-specific circRNA expression profiles, with differences in both the nature and abundance of circular transcripts.

### 2.3. EC-Specific circRNA Identification

To identify potential circRNA expression signatures, we cross-referenced the lists of circRNAs expressed in each cell type and plotted the results in a Venn diagram using the jvenn platform [50] (Figure 4A,B and Appendix A). We found variable proportions of circRNAs detected in only one cell type and some common circRNAs, within ECs and non-ECs. For endothelial cells, HUVECs have only 1.24% exclusive circRNAs, HDLECs have 11.05%, HUAECs have 52.74%, and HCAECs have 53.14% (Appendix A). We only found four circRNAs specific to endothelial cells (Figure 4A,B). We processed the gene lists associated with these cell-type-specific circular transcripts using the online gene set analysis tool called Enrichr (https://doi.org/10.1002/cpz1.90 (accessed on 1 September 2023)). The aim here was to identify possible signalling pathways or the biological processes significantly more represented in each gene set, looking for patterns.

For the very few circRNAs exclusive to HUVECs, only six in total, it is difficult to obtain significant results. Nevertheless, one relevant signalling pathway caught our attention: the integrated breast cancer pathway (WP1984), with a *p*-value of 0.059 in association with the circRNA circZMYND8_A, formed by the circularisation of exons 20 and 21 of the ZMYND8 gene (also called RACK7). Although the ZMYND8 mRNA is expressed at high levels in all the other endothelial cells, circZMYND8_A is only found in HUVECs and among the top 40% most expressed in these cells. For HDLECs, the endothelial cell differentiation pathway is significantly overrepresented through circPROX1 and circBMPR2, respectively, formed by the circularisation of exons 2 to 4 of the PROX1 and exon 12 of the BMPR2 gene. If the PROX1 mRNA is expressed at a very low level in other ECs, compared to HDLECs, none of them express any circular isoform for this gene. For HUAECs, the DNA damage repair pathway is significantly represented in the cell-specific circRNA gene set with a particularly high number of associated genes (51 in total), including the histone methyltransferase SETD2, the breast-cancer-associated protein BRCA2. For HCAECs, the regulation of platelet aggregation is overrepresented, with a particular interest in the plasminogen activation inhibitor SERPINE1. The circRNA is formed by the circularisation of exons 4 to 7. Although the SERPINE1 mRNA is expressed in all ECs, at a lesser level in the three others, HCAECs are the only ones producing circular transcripts for this gene and circSERPINE1 is the 4th highly expressed circRNA detected in these cells.

All these results highlight the fact that each endothelial cell expresses a specific set of circular RNAs, notwithstanding the expression level of the matched linear RNAs. The exploration of the function of these circRNAs has the potential to shed light on new ways to target and modulate biological functions of endothelial cells in pathological contexts.

We were particularly interested in the four circRNAs that are common to HCAECs, HDLECs, HUAECs and HUVECs, and absent from our non-EC controls: circCARD6, circPLXNA2, circCASC15 and circEPHB4 (Figure 4B). CircCARD6 is generated by the circularisation of exon 3 of the Caspase Recruitment Domain Family Member 6 (CARD6) gene that encodes NF-κB activators, which are implicated in inflammatory bowel diseases and gastrointestinal cancers [51,52]. CircPLXNA2 is produced by the backsplicing of exon 2 and exon 3 of the Plexin A2 (PLXNA2) gene that encodes a transmembrane coreceptor for semaphorins 3A and 6A. The function of CircPLXNA2 has only recently been studied in myoblasts, in which the circRNA promotes proliferation and inhibits apoptosis by sponging miR-12207-5p, thus maintaining the expression of the P53 tumour suppressor inhibitor MDM4 [53]. CircCASC15 is generated by the circularisation of exon 7 of the Cancer Susceptibility Candidate 15 (CASC15) long non-coding RNA. Though circCASC15 has previously been abundantly detected in melanoma cell lines [54], its functions have not been evaluated, neither in tumour cells nor in endothelial cells. The last endothelial-specific circRNA circEPHB4 is produced by the backsplicing of exons 5 and 6 of the Ephrin Type-B Receptor 4 (EPHB4) gene that codes a transmembrane protein playing an essential role in vascular development by binding to its ligand ephrin-B2. The function of CircEPHB4 has not been investigated yet. The EC-specific circRNAs are, on the one hand, significantly associated with the endothelial migration pathways linked to angiogenesis (Figure 4C, Appendix A) through circPLXNA2 and circEPHB4, and, on the other hand, associated with the promotion of cancer progression through circCARD6 and circCASC15.

All these results are testimonies of the pool of relevant regulatory molecules represented by circular RNAs for each endothelial cell; as a group, they help to explore and find new molecular markers and (lymph)angiogenesis modulators in both physiologic and cancer-associated pathological conditions (for summary see Appendix A).

### 2.4. Specific circRNA Expression Validation by RT-qPCR and RNase R Treatment

To validate the observations made on the RNA sequencing data, we selected a list of specific and shared circular RNAs to be quantified by RT-qPCR based on those that display the highest read count for each cell type to maximize the chances of detection. We established a list of specific and common circRNAs, as displayed in Table 1 with their annotations. A graphical representation of these circRNA structures is available in Appendix A.

For this list of circRNAs, as for the global analysis above, there was no significant correlation between the expression level of specific circRNAs and the expression level of the matched linear transcripts in the RNAseq data (Figure 5A,B). This was confirmed by the comparison of the coefficients of variance between the two transcript types (Figure 5C). The RT-qPCR quantifications showed relative levels of circRNAs that were consistent with the RNAseq expression patterns for 11 circRNAs out the 17 quantified, i.e., a 65% validation. This small discrepancy between the RNAseq and RT-qPCR expression profiles is not uncommon, as it has been shown to be caused by a strong RNA secondary structure, as well as sample preparation methods that affect RNAseq quantification for some transcripts [55]. In addition, RT-qPCR enables the detection of a given circular RNA via the specific amplification of the backspliced junction, whereas in RNAseq data, circular RNAs are identified via the detection of reads overlapping the backspliced junction. For lowly expressed circRNAs, the lower probability of reads overlapping the backspliced junction can lead to an underestimation of their abundance in RNAseq compared to RT-qPCR quantification. Nevertheless, the circRNAs among the 65% validated expression profiles are very robust candidates for further analysis, particularly circPLXNA2, which is common to all ECs. The linear RNA RT-qPCR expression patterns, on the other hand, are drastically less successfully validated, with only 2 out 17, i.e., a 12% validation. These differences can explain how the RT-qPCR data showed no significant difference between the circRNA and the matched linear RNA coefficient of variance, in contrast to the RNAseq data. The detailed tables of circRNA read counts and RT-qPCR quantification values are, respectively, in Appendix A.

We also sought to confirm that the circRNAs of interest are *bona fide* circular transcripts by testing their resistance to the exonuclease RNase R. Linear RNAs are more sensitive to RNase R degradation than circRNAs for short digestion times. Here, we quantified the level of circular vs. matched linear transcripts for ECs using RT-qPCR. The results are normalized using the untreated condition for each transcript and presented in Figure 6A for the common circRNAs between ECs, and in Figure 6B for the cell-type-specific circRNAs (Figure 6A,B). We see that circular RNAs are systematically more resistant to RNase R degradation than their linear counterpart in all endothelial cells, except for HCAEC-specific circRNAs, for which we did not obtain usable results. This demonstrates that the selected transcripts are indeed circular RNAs.

In the end, we were able to validate most of the relevant specific circRNAs listed for ECs in our conditions, which makes them suitable candidates for future functional studies.

## 3. Discussion

Circular RNAs are covalently closed RNA molecules that are abundantly and endogenously produced in eukaryotic cells, with cell-type-specific expression patterns and an undoubted regulatory potential in gene expression [29,31,54]. A growing body of research describes the large-scale involvement of circRNAs in the regulation of gene expression in physiology and physiopathology. Many circRNAs regulate cell proliferation, migration and invasiveness, and act positively or negatively on tumour progression, either through miRNA/RBP sponging mechanisms or by being translated into a tumour suppressor/promoter peptide [56,57,58]. The impact of tumour-cell-derived circRNAs on the surrounding vascular function has been touched upon, as illustrated by the circEHBP1/miR-130a-3p/TGFbR1 axis [46]. However, the role of circRNAs produced by endothelial cells (ECs) themselves remains elusive. Functional differences between arteries, veins and lymphatic vessels are vastly defined by the differential gene expression profiles within arterial, veinous, and lymphatic ECs. Circular RNAs participate in this landscape of cell specification and cell response mechanisms. Based on embryonic origins, we could expect more similarities between veinous and lymphatic ECs, compared to arterial ECs, for example.

In this study, we aimed to explore the diverse landscape of circRNA expression in ECs, identifying potential new molecular signatures and EC-specific markers, as preliminary work for further functional analysis.

### 3.1. Endothelial Cell Specific Expression Profiles

Our analysis reveals the existence of EC-specific circRNA expression profiles, with differences in both the nature and abundance of circular transcripts. We find that within ECs, HUAECs express the most circRNA isoforms, followed by HDLECs, HUVECs and HCAECs; these have, respectively, 19.48%, 6.05%, 4.92%, and 2.96% circular transcripts. Knowing that circRNAs are relatively stable once produced, one might speculate that the cell generation time might influence the amount of circular RNAs present in the cells (i.e., an increased cell generation time might favour the accumulation of circular RNAs). However, this does not appear to be the case here, as the cells used in this study have similar generation times: 46.1 h for HUAEC, 47 h for HUVEC, 43.1 h for HCAEC [59], 46 h for HDLEC (personal data), 47.7 h for SMC-UA [60], and 40–60 h for SAT-SVF [61]. The higher proportion of circRNA transcripts in HUAECs could also reflect an increased level of backsplicing in these cells compared to the other ECs. Setting aside HCAECs for their sequencing depth bias here, when we look at the backsplicing rate at each circular RNA-producing locus, it appears that HUVECs have the highest rate. The circRNA-producing genes in HUVECS generate 31% circular transcripts on average, compared to 20% for HUAECs and 14% for HDLECs. This implies that, for HUVECs that globally produce fewer circRNA molecules than the two other ECs, backsplicing is enhanced only at restricted loci in the genome. On the other hand, HUAECs have a greater number of circRNA-producing genes, while backsplicing is less prominent for each of them, hinting at cell-specific regulations. In addition, we can see that the number of circRNAs produced by a given locus is mainly independent of its transcription rate in ECs, which points towards specific backsplicing regulation mechanisms in ECs. One of the documented backsplicing modulators is the intervention of RNA Binding Proteins (RBPs). For example, the RNA-editing enzymes ADARs and their cofactor, the RNA helicase DHX9, negatively regulate circular RNA biogenesis [62,63], while other RBPs such as QKI [64], FUS [65], hnRNPL [66], hnRNPM [67], RBM20 [68], Mbl [69], and ILF3/NF90-NF110 [70] positively regulate this process. It would be interesting to measure the level of correlation between the circRNA expression profile in each cell type studied here and the respective protein levels for the different RBPs. Another instructive parameter to assess would be the RNA polymerase II transcription speed at the circRNA-producing loci across ECs, using global run-on sequencing (GRO-seq), for example, since it has been shown to regulate alternative splicing, including backsplicing [71]. Thus, we could gain a better understanding of the factors influencing circular RNA biogenesis in ECs.

### 3.2. Endothelial Cell Specific Circular RNAs

Our results reflect the fact that each endothelial cell expresses a specific set of circular RNAs that are absent in the other ECs and produced by important genes involved in cell survival, proliferation, migration, invasiveness and angiogenic processes. In the case of HUVECs, for example, circular RNA circZMYND8_A is produced from the gene encoding a zinc finger protein described as an anti-metastatic molecule, acting as a co-repressor of genes involved in cell invasiveness; this is via histone modification reader activity [72,73]. In the case of HUAECs, there are 51 specific circular RNA-producing genes, whose function is linked to the DNA damage repair pathway. These include the histone methyltransferase SETD2, involved in DNA repair via homologous end joining recombination, and multiple excision repair proteins (ERCCs). All these genes code tumour suppressors, and their mRNAs are expressed ubiquitously in contrast to the circular isoforms. For example, a report published earlier this year described a circular isoform of SETD2 that was shown to promote the proliferation and invasion of trophoblasts, and to inhibit their apoptosis in a rat model of preeclampsia [74]. If this property is transposable in arterial endothelial cells, we can speculate that circSETD2 could regulate the proliferation of arterial ECs and thus arterial wall morphology and potentially angiogenesis.

Interestingly, the specific circRNA-producing genes are in part associated with cell-specific functions. This is the case for PROX1 in HDLECs. PROX1 encodes a transcription factor that is important for the formation of various tissues during development and is essential for the specification of lymphatic endothelial cells (LECs) during development and phenotype maintenance in adult vertebrates [75,76]. PROX1 expression dysregulation in LECs causes lymphatic vasculature malformations that lead to lymphedema and are associated with obesity and metabolic diseases [77]. The PROX1 transcription factor is responsible for the expression of the vascular endothelial growth factor receptor 3 (VEGFR3), which is stimulated by tumour-cell-derived VEGFC; this triggers tumour lymphangiogenesis and establishes the main route of dissemination for solid tumours [14,78,79,80]. The possibility that the PROX1 circular RNA could display even a fraction of the associated transcription factor’s impact on the regulation of gene expression in LECs is enough to justify its thorough characterisation in the context of tumour lymphangiogenesis, lymphedema, and other lymphatic disorders.

Another example is the SERPINE1 gene in HCAECs. The plasminogen activation inhibitor SERPINE1 is important in the prevention of persistent blood clots and is thus involved in the prevention of arterial thrombosis. The expression dysregulation of such genes is linked to multiple pathological conditions, including vascular dysfunctions like lymphedema and thrombosis, but also cancer formation and metastasis.

None of these circRNAs have ever been studied in the context of endothelial cell function, although it seems relevant to do so as they have the potential to shed light on new ways to target and modulate the biological functions of endothelial cells in pathological conditions.

### 3.3. Specific Endothelial Cell circRNA Signature

Lastly, we identified a specific signature of four circRNAs only detected in ECs: circCARD6, circPLXNA2, circCASC15 and circEPHB4. Among them, circEPHB4 and circPLXNA2 are both derived from genes encoding endothelial cell adhesion, migration, and guidance molecules, making them attractive targets for the study of vascular physiology and associated pathologies. Regarding the EPHB4 gene, it encodes a guidance molecule involved in endothelial cell progenitor migration and differentiation, and defects in its expression can cause vascular malformations in arteries, veins, and lymphatic vessels [81,82,83]. In tumour vessel xenografts, EPHB4 expression switches the vascularization program from sprouting angiogenesis to circumferential vessel growth, and reduces the permeability of the tumour vascular system via the activation of the angiopoietin-1/Tie2 system at the endothelium/pericyte interface. CircEPHB4, as defined here, has not been investigated yet, but another circRNA from the same locus (hsa_circ_0081519) has been shown to positively regulate stemness and the proliferation of glioma cells by sponging miR-637 and up-regulating SOX10. Since many circRNAs exhibit opposite properties to those of their parental gene, we could speculate that circEPHB4 favours sprouting angiogenesis by supporting EC proliferation and may be of interest in tumour metastasis, for example. Meanwhile, the PLXNA2 gene encodes a transmembrane coreceptor for semaphorins 3A and 6A. Semaphorins are essential extracellular signalling proteins for the development and maintenance of many organs and tissues. They function as regulators of morphology and motility in many different cell types, including those that make up the nervous, cardiovascular, and immune system, as well as in cancer cells [84]. A high expression of PLXNA2 has been associated with invasive breast ductal carcinoma compared to benign tumours [85], while in ECs, the co-receptor acts as an antiangiogenic and pro-permeability factor through semaphorin 3A signalling. It is also capable of normalizing tumour-associated vasculature in interaction with VEGF signalling [86]. Given the functions recently characterized for circPLXNA2 in proliferation and apoptosis [53], it is reasonable to assume that circPLXNA2 has great potential as a regulator of angiogenesis and permeability in endothelial cells, its positive or negative effect depending on which signalling pathway the circular RNA will take part in.

The other two, circCARD6 and circCASC15, are produced by genes more closely associated with tumour progression, as they have been linked to the epithelial–mesenchymal transition (EMT). CircCARD6 is produced from the CARD6 gene, which encodes an NF-κB activator; this is implicated in inflammatory bowel diseases and gastrointestinal cancers [51,52]. The NF-κB transcription factor targets genes involved in the progression of inflammation by triggering cell survival, proliferation, inflammation response, angiogenesis, cell adhesion, invasion, and metastasis, either directly in the cell or via the secretion of cytokines and chemokines [87]. CircCARD6 has been recently studied in posterior capsular opacification (PCO) characterized by an epithelial-to-mesenchymal transition (EMT) of human lens epithelial cells after surgery [88]. This circular RNA has been shown to increase FGF7 expression in lens cells by sponging miR-31 and promoting proliferation, metastasis and EMT, thus suggesting that circCARD6 is a potential target for the treatment of PCO. Interestingly, miR-31 is an important factor in the specification of ECs during vasculogenesis through the inhibition of PROX1 expression in blood vascular ECs, thus inhibiting the lymphatic fate [89]. Hence, circCARD6 could participate in the fate determination and/or maintenance of ECs, particularly in lymphatic ECs, in which they might also sponge miR-31 to maintain PROX1 expression. Regarding circCASC15, it is generated by the CASC15 gene, which produces a long non-coding RNA that facilitates tumour cell proliferation and EMT in vitro but also promotes melanoma and gastric cancer progression in patients [90,91]. The lncRNA CASC15 promotes proliferation by acting on cell cycle advancement and EMT via a competitive binding to miR-33a-5p, thus protecting the transforming transcription factor ZEB1 mRNA. The lncRNA can also positively regulate the expression of the adjacent oncogene SOX4 [92]. Although nothing is known about circCASC15, we can speculate that this circRNA could harbour properties that are similar to its linear counterpart, since the lncRNA regulates gene expression through the sequence-specific binding of partners. This could suggest that, when expressed by endothelial cells in a tumoral context, these circRNAs participate in the EMT of these cells, inducing the formation of cancer-associated fibroblasts that form part of the tumour stroma and in turn favour EMT for tumour cells via cytokine secretion.

Mechanistically, we can speculate that EC-specific circRNAs act as competing miR sponges, as previously described in the literature in other cellular contexts for circCARD6 [88], circPLXNA2 [53] and circEPHB4 [93]. They could similarly serve as competing RBP traps or be translated into alternative protein isoforms, acting independently or not of the main protein product of the parental gene.

### 3.4. Biomedical Implications

Circular RNAs present great potential as biomarkers due to their stability and specific expression under different physiological and pathological conditions. Another important clinical consideration in the use of EC-specific circRNAs as biomarkers is their accessibility. Indeed, endothelial cells are in direct contact with the fluids routinely withdrawn for analysis (blood or lymph) and are known to abundantly produce circRNA-containing exosomes, which are found in these fluids [45,94,95,96]. The small vesicles play an essential role in intercellular communication, by transferring proteins, RNAs (messenger RNAs, microRNAs, and circRNAs [96]) and other bioactive molecules between cells. Exosomal circRNAs have been shown to be able to modify host cell behavior and either promote or inhibit cancer progression by targeting both other cancer cells and the microenvironment in multiple cancer settings [97]. Although no EC-derived exosomal circRNA has been characterized yet, a study by Li et al. in 2018 showed that pancreatic carcinoma cells could transfer the circRNA circ-IARS to HUVECs, thus increasing their permeability and promoting invasion and metastasis through a miR-122/RhoA axis [98]. Interestingly, exosome-transmitted circCARD6 in lens epithelial cells has recently been associated with a vision-disrupting complication of cataract surgery [88]. It would be very instructive to see whether the complete or partial list of endothelial-specific circRNAs can be detected in exosomes produced from the ECs used in this study, under normal and pathological conditions in patients.

CircRNAs with validated expression profiles may be utilized in diagnostic tools, offering a novel approach for assessing endothelial cell health or identifying specific vascular disorders. One example is a common pathology such as cancer, which often has a poor prognosis and is costly disease to treat. We know that the later cancer is diagnosed, the worse the prognosis. It is therefore essential to have early biomarkers, not only for medical reasons, but also for economic ones. A critical step in the development of primary solid tumors is the process of angiogenesis and lymphangiogenesis. Finding highly sensitive and specific early markers, such as the circular RNAs secreted by ECs during these processes, could undoubtedly bring benefits in terms of early diagnosis and hence future disease progression. A similar concept can be applied to vascular pathologies in general, which are the leading cause of death in industrialized countries. Here also, finding specific circRNA markers secreted by endothelial cells for these pathologies could be of real benefit. One can also speculate that tailoring medical interventions based on the circRNA profiles of individual patients could lead to more effective and personalized treatment strategies. Thus, circulating circRNAs may be used for non-invasive diagnosis or the monitoring of specific vascular diseases, cardio-vascular diseases, cancers, and other illnesses, or to assess therapeutic responses in different pathologies.

Therapeutically, the use of circRNAs may also be considered. Thus, the circRNAs involved in cancer-related pathways could be explored as therapeutic targets for inhibiting tumor growth and metastasis. It can also be suggested that the circRNAs identified as crucial in endothelial cells may become targets for drug development, potentially leading to the creation of novel therapeutics for vascular and cancer-related disorders. RNA-based targeted therapies could be developed by producing and administering circRNA-containing vesicles to patients. Conversely, if the expression of EC-specific circRNAs is deleterious, targeted degradation via the administration of specific microRNAs could be a therapeutic strategy.

### 3.5. Conclusions

To conclude, our work on the characterization of EC circRNAs reveals a remarkable distinction: circular RNAs exhibit a strikingly unique expression profile that is specific to individual endothelial cell types compared to linear RNAs from the same cell type. This specificity positions them as a superior tool for endothelial cell type distinction and identification. As a result, we unveiled distinct circRNA signatures unique to various endothelial cell types. Notably, we identified a precise set of circRNAs that are common to all EC types tested in our study, including circCARD6, circPLXNA2, circCASC15 and circEPHB4. These circRNAs are associated with the genes influencing endothelial cell migration pathways and cancer progression. The in-depth study of their functionalities promises to unravel crucial insights into both the physiological and pathological aspects of (lymph)angiogenesis. Furthermore, our work paves the way for the use of endothelial-cell-specific circRNAs as diagnostic biomarkers if they are proven to be secreted, or even as therapeutic targets in patients with vascular pathologies or those with vascular consequences such as cancer. However, it is important to note that further research and validation are necessary to establish the clinical significance and applications of these findings.

## 4. Materials and Methods

### 4.1. RNA Sequencing

*Total RNA extraction: Total RNA was extracted using a GeneElute Total RNA purification kit (RNB100, Sigma-Aldrich, Saint-Louis, MO, USA).

*RNA QC: Sample quality control and libraries preparation were performed at the GeT-Santé facility (Inserm, Toulouse, France, get.genotoul.fr). The RNA concentration and purity were determined using a ND-2000 Spectrophotometer (Thermo Fisher Scientific, Waltham, MA, USA). The integrity of RNA was checked with a Fragment Analyzer (Agilent Technologies, Santa Clara, CA, USA), using the RNA Standard Sensitivity Kit. The 260/280 purity ratios were all ≥1.8, and the integrity indices revealed good values (9.6–10 RIN and >1.8 28S/18S ratios).

*Libraries preparation: RNA-seq paired-end libraries were prepared according to Illumina’s protocol with some adjustments, using the TruSeq Stranded Total RNA Gold library prep Kit (Illumina, San Diego, CA, USA). Briefly, for each sample, 1000 ng of total RNA was first Ribo-zero depleted using Illumina Ribo-Zero probes. Then, the remaining RNA was fragmented during 3′ and retrotranscribed to generate double-stranded cDNA. Compatible adaptors were ligated, allowing the barcoding of the samples with unique dual indices. Then, 10 cycles of PCR were applied to amplify the libraries, and an additional final purification step allowed 280–700 pb fragments to be obtained.

The quality of the libraries was assessed using the HS NGS kit on the Fragment Analyzer (Agilent Technologies, Santa Clara, CA, USA).

*Libraries quantification: Libraries quantification and sequencing were performed at the GeT-PlaGe core facility (INRAE, Toulouse, France). Libraries were quantified by qPCR using the KAPA Library Quantification Kit (Roche, Basel, Switzerland) to obtain an accurate quantification.

*Sequencing: Libraries were equimolarly pooled and RNA sequencing was then performed on 3 lanes of the Illumina HiSeq 3000/4000 instrument (Illumina, San Diego, CA, USA), using a paired-end read length of 2 × 150 pb with the appropriate Illumina HiSeq sequencing kits. Between 90 and 119 million paired-end raw reads were produced per sample.

### 4.2. RNA Sequencing Dataset Collection and Analysis

Total paired-end RNA sequencing data were retrieved from the NCBI publicly available Sequence Read Archive for Human Artery Endothelial Cells (HUAECs), Human Umbilical Vein Endothelial Cells (HUVECs), Human Coronary Artery endothelial Cells (HCAECs), Smooth Muscle Cells from Umbilical Artery (SMC-UAs), and Superficial Adipose Tissue Stromal Vascular Fraction (SAT-SVFs). Original paired-end RNA sequencing data were used for Human Dermal Lymphatic Endothelial Cells (HDLECs). See Table 2 below for data sources.

The reads coming from the different samples were first pre-processed using Fastp [99] to trim adaptors and low-quality reads. The remaining reads were then aligned to the human hg38 reference genome with BWA-MEM [100]. The de novo detection of circular RNAs was performed using the CIRI2 v2.0.6 software [47]. Only circRNAs with 10 or more reads were analysed and compared. The Jvenn [50] platform was used to create Venn diagrams.

### 4.3. RNA Sample Recovery for RT-qPCR

Total RNAs were extracted from primary cultures of HDLECs (ref. C-12216, PromoCell, Heidelberg, Germany) using TRIzol Reagent^TM^ from Invitrogen, according to the supplier’s instructions. The total RNA extracts from HUVEC and SAT-SVF were provided, respectively, by the Lenfant and Bouloumier’s teams in the I2MC of Toulouse, France. The total RNA extracts were purchased from PromoCell for HUAECs (ref.c-14013) and from Tebubio for HCAECs (ref. P00057781) and SMC-UAs (ref. P00057777). All RNA samples were stored at −80 °C before use.

### 4.4. Ribonuclease R (RNase R) Treatment

For each condition, 1 µg of total RNA was incubated at 37 °C for 15 min in the presence of 3U of RNase R (Lucigen^®^, LGC, Teddington, Middlesex, UK) or without for the control conditions, followed by a 3 min inactivation step at 95 °C.

### 4.5. RNA Quantification by RT-qPCR Analysis

The cDNAs were synthesised using 1 μg of treated RNase R or the control total RNAs from each sample using the High-Capacity cDNA Reverse Transcription kit (Applied Biosystems^®^, Waltham, MA, USA), according to the supplier’s protocol. Quantification by qPCR was performed on cDNA samples using the ONEGreen FAST qPCR Premix (Ozyme^®^, Saint-Cyr-l’École, France) with the primer pairs listed in the Table 3 below, on a ViiA7^TM^ instrument (Applied Biosystems^®^, Waltham, MA, USA). The primers used were purchased from Integrated DNA Technologies^®^, Coralville, IO, USA.

### 4.6. Statistical Analysis

Statistical significance for comparisons of the mean coefficient of variance and circ/lin ratio in Figure 2 and Appendix A, comparisons of the mean transcripts count in Appendix A, and the comparison of the mean coefficient of variance in Figure 5 were, respectively, assessed using two-way Anova, ordinary one-way Anova, and Student’s *t*-test. Statistical details and error bars are defined in each figure legend: *p* < 0.0001 (****), *p* < 0.001 (***), *p* < 0.01 (**) and *p* < 0.05 (*).

## Figures and Tables

**Figure 1 ijms-25-00680-f001:**
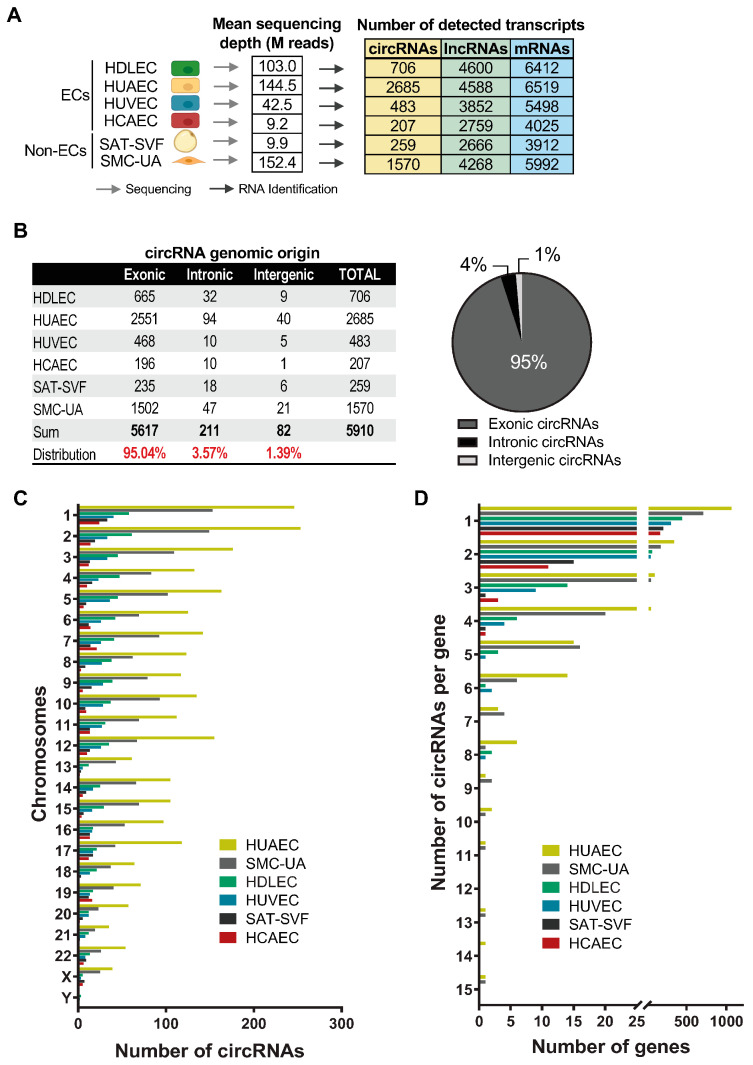
Extended Landscapes of circRNAs in Endothelial Cells (ECs). (**A**) RNA sequencing data overview. Only circRNAs with 10 or more reads are accounted. (**B**) Number of distinct circRNAs sorted by genomic origin for each cell type (left) and the global proportion of exonic, intronic and intergenic circRNAs for all cell types combined (right). (**C**) Number of distinct circRNAs expressed per chromosome for each cell type. (**D**) Number of distinct circRNAs expressed per gene for each cell type.

**Figure 2 ijms-25-00680-f002:**
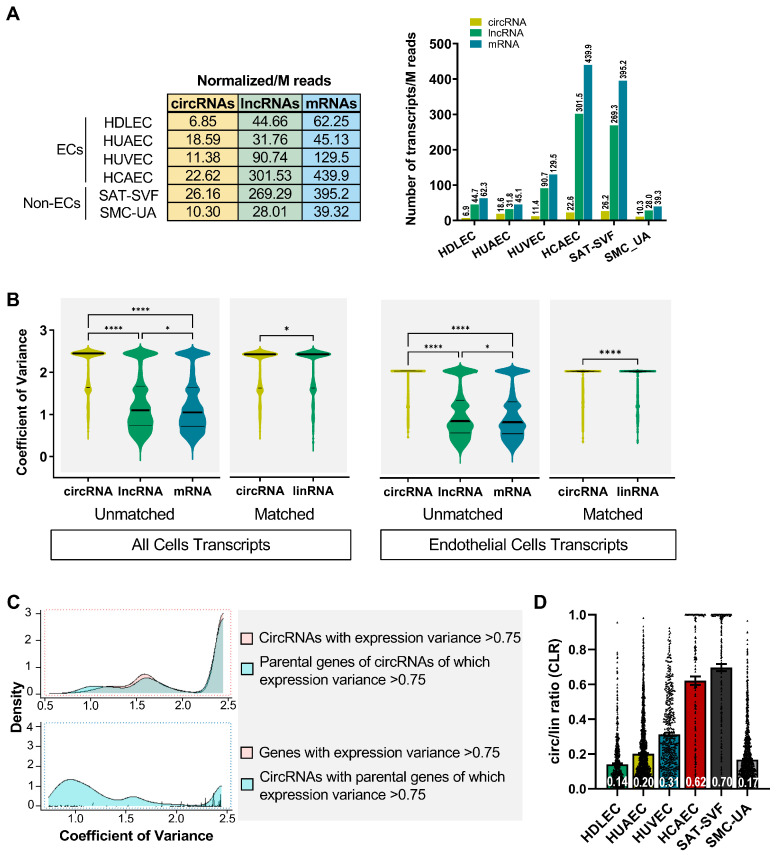
The nature and abundance of circRNAs varies between ECs. (**A**) Different transcript read counts normalized by million (M) of total reads for each cell type. (**B**) CircRNAs, lncRNAs and mRNAs independent expression variance coefficient across all cell types or across ECs, and matched circRNA/linear RNA expression variance coefficient. Ordinary one-way ANOVA comparison *p* values are plotted on the graphs: * <0.05; **** <0.0001. (**C**) Density plot of circRNAs with expression variance over 0.75 and matched linear RNA variance from the same gene (red dotted rectangle). Density plot of linear RNAs with expression variance over 0.75 and matched circRNA variance from the same gene (blue dotted rectangle). (**D**) Circular on linear transcript read counts for circRNA-expressing genes (mean in white digits). All differences are significant by two-way ANOVA comparison.

**Figure 3 ijms-25-00680-f003:**
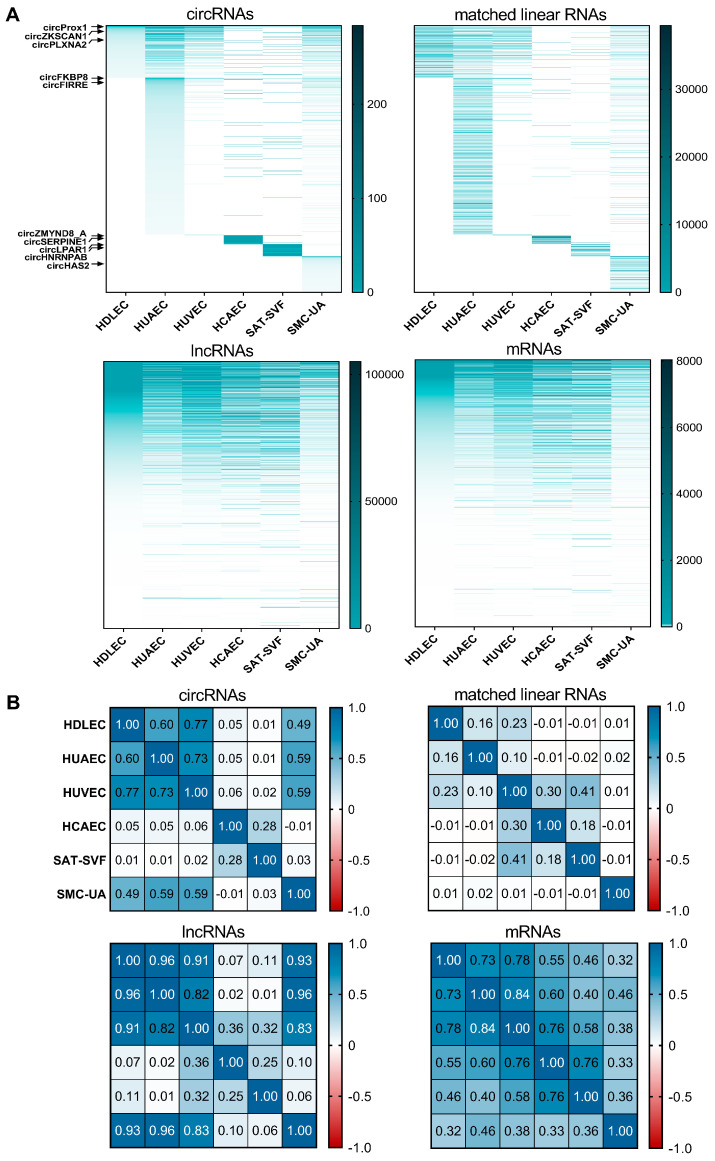
circRNAs are better at distinguishing between ECs than mRNAs. (**A**) Heatmap representing expression levels for circRNA and matched linear RNA from the same gene, all noncoding RNAs and mRNAs. Relevant circRNA positions are indicated by black arrows. (**B**) Corresponding Pearson correlation matrixes. *p* values are presented in Appendix A.

**Figure 4 ijms-25-00680-f004:**
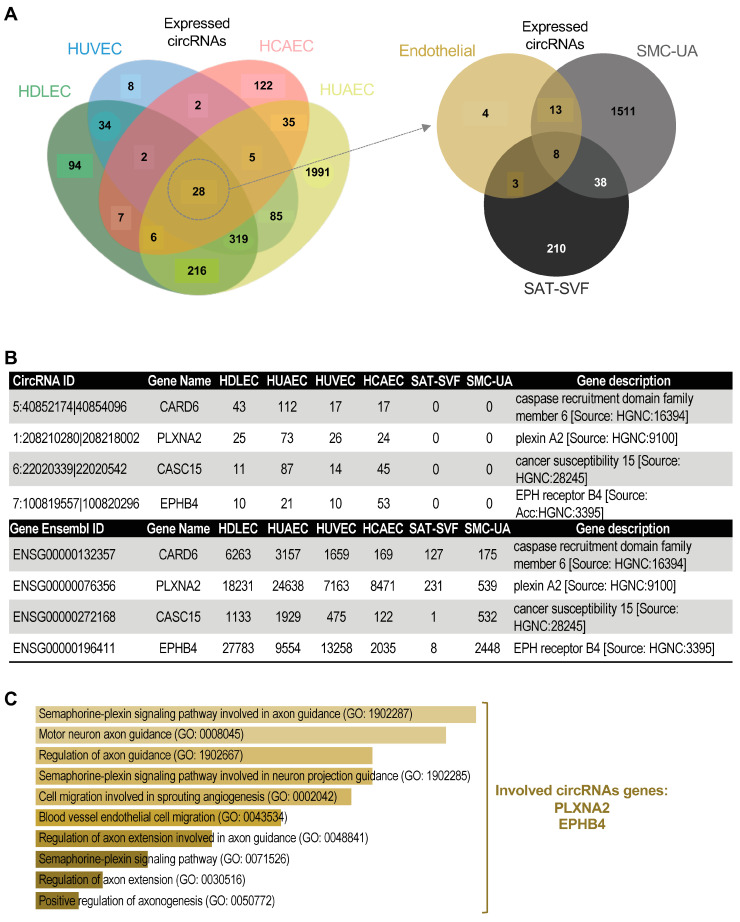
There are cell-type-specific circRNAs. (**A**) Venn diagram representing the comparison of the lists of circRNAs expressed in all ECs (left) and the crossing between non-EC circRNAs and the 28 circRNAs common to ECs. (**B**) Description of the four EC-specific circRNAs (top) and their linear counterpart (bottom), with the read count in each cell type and gene description. (**C**) Enrichr gene ontology analysis for EC-specific circRNAs.

**Figure 5 ijms-25-00680-f005:**
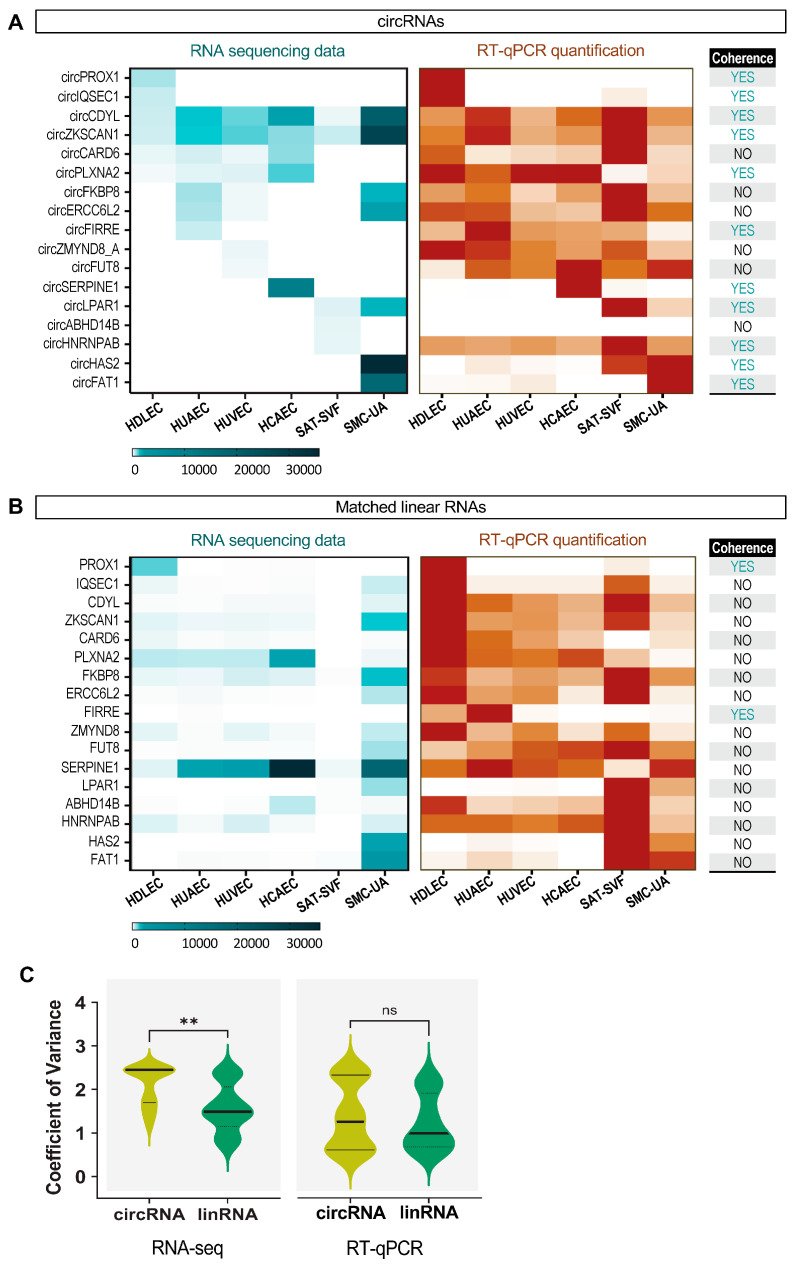
Validation of specific circRNAs expression by RT-qPCR. (**A**) Heatmap representing RNAseq expression levels for selected circRNAs and corresponding qPCR quantifications. (**B**) Heatmap representing RNAseq expression levels for selected circRNA linear counterparts and corresponding qPCR quantifications. (**C**) Expression coefficient of variance for selected circRNAs and linear counterparts across cell types from RNAseq data and qPCR quantifications. *T* test comparisons *p* values are plotted: ** <0.01. ns: not significant.

**Figure 6 ijms-25-00680-f006:**
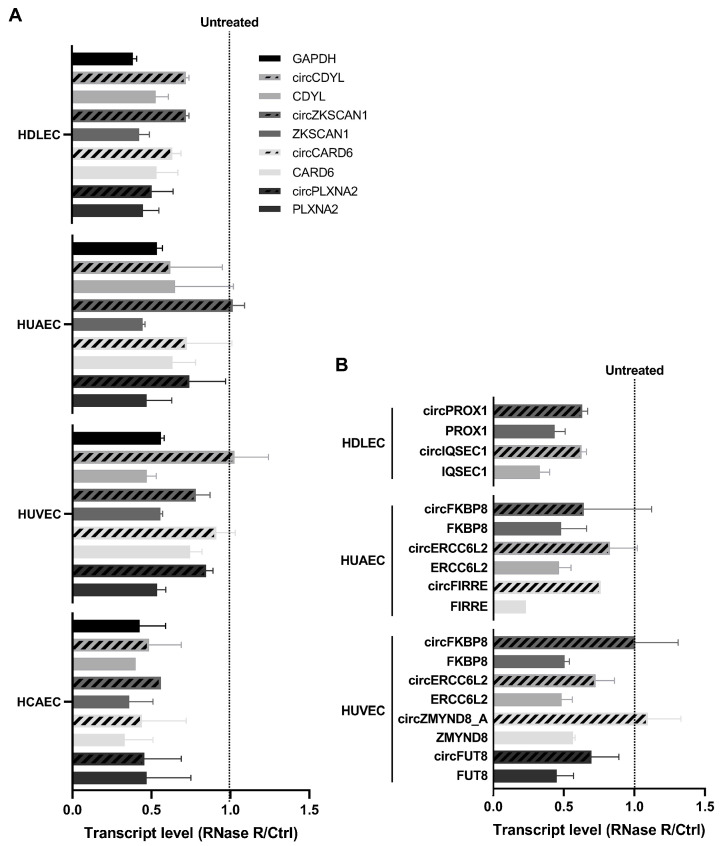
RNase R treatment validation of EC-specific circRNAs. (**A**) Quantification of circRNAs that are common to all cell types and their linear counterparts via RT-qPCR following RNase R treatment. The results are normalized using the untreated control for each transcript. (**B**) Quantification of cell-type-specific circRNAs and their linear counterparts via RT-qPCR following RNase R treatment. The results are normalized using the untreated control for each transcript. The experiment was repeated twice independently.

**Table 1 ijms-25-00680-t001:** Selected specific circRNAs and their annotations.

circBase ID	Gene Name	circRNA Name	circRNA Localization	Strand	Origin	Genomic Lenght (pb)	Spliced Lenght (nt)	Gene Description [Sources]	Gene Associated Pathologies	Known circRNA Functions	Sources (Pubmed ID)
hsa_circ_0111 952	PROX1	circPROX1	chr1:21399646 9-214011715	+	exonic	15247	2095	prospero homeobox 1_protein coding transcription factor activity-cell specification [HGNC: 9459; NCBI: 5629]	lymphatic malfunctions; cancers; obesity	none	19056879; 16170315 28024299; 22023334
#N/A	IQSEC1	circlQSEC1	chr3:13129305 -13129525	−	intronic	221	221	IQ Motif And Sec7 Domain ArtGEF 1 _protein coding GEF activity-endosomal protein traffic [HGNC: 29112 NCBI: 9922]	intellectual development disorder	none	31607425
hsa_dic_0008 285	CDYL	circCDYL	chr6:4891713- 4892379	−	exonic	667	667	Chromodomain Y Like_protein coding histone modification- transcriptional repression [HGNC: 1811; NCBI: 9425]	cancer chemoresistance	pro-myocardial regeneration in vitro	19061646;31367252 32522972
hsa_circ_0001 727	ZKSCAN1	circZKSCAN1	chr7:10002341 9-100024307	+	exonic	889	668	Zinc Finger With KRAB And SCAN Domains 1 _protein coding transcription factor activity-[HGNC: 13101; NCBI 7586]	gastric cancer, Hepatocellular cancer	anti-hepatocellular cancers progression	7557990; 28211215 33439397
hsa_dirc_0005 895	CARD6	circCARD6	chr5:40852174 40854096	+	exonic	1923	1923	Caspase Recruitment Domain Family Member 6 protein coding [HGNC: 16394; NCBI: 84674]	inflamatory bowel diseases; pro- intestinal cancers	pro-posterior capsul opacification (eye)	12775719; 16418290 20025480; 33844960,
hsa_drc_0002 472	PLXNA2	circPLXNA2	chr1:20821028 0-208218002	−	exonic	7723	1451	Plexin A2 protein coding [HGNC: 9100 NCBI 5362]	axonal guidance disfunctions Breast cancer tumorigenesis	pro-profiferation and anti-apoptotic in myoblasts	16402134;21925246 10.3390ljjms24065459
hsa_drc_0000 914	FKBP8	circFKBP8	chr19:1853760 1-18538436	−	exonic	836	394	FKBP Proly Isomerase 8 _protein coding [HGNC: 3724; NCBI: 23770]	spina bifida	none	18003640; 32969478
#N/A	ERCC6L2	circERCO6L2	chr9:95978061 -96004701	+	exonic	26641	337	ERCC Excision Repair 6 Like 2 _protein coding [HGNC: 26922 NCBI 375748]	bone marrow failure syndrome	none	4507776
#N/A	FIRRE	circFIRRE	chrX:13174930 6-131794466	−	exonic	45161	872	Firre Intergenic Repeating RNA Element_IncRNA [HGNC: 49627; NCBI 286467]	various cancers	chrX inactivation and nuclear organisation? RNA stability?	35110535;3 35988459 29678151;30124921;
hsa_dirc_0007 026	ZMYND8	dircZMYND8_A	chr20:4726228 8-47276795	−	exonic	14508	623	Zinc Finger MYND-Type Containing 8 protein coding [HGNC: 9397; NCBI: 23613]	various cancers	none	11003709; 27477906
hsa_drc_0003 028	FUT8	circFUT8	chr14:6556133 7-65561766	+	exonic	430	430	Fucosyltransferase 8 _protein coding [HGNC: 4019 NCBI: 2530]	various cancers; congenital disorders	cancer suppression or promotion?	19302290; 26289314; 29304374;32072011; 33500381
#N/A	SERPINE 1	circSERPINE1	chr7:10113191 6-101135801	+	exonic	3886	541	Serpin Family E Member 1_protein coding [HGNC: 8583 NCBI: 5054]	Thrombosis PAI1-deficiency	none	3922531; 9207454
hsa_dirc_0087 960	LPAR1	airdLPAR1	chr9:11097207 3-110973558	−	exonic	1486	226	Lysophosphatidic Receptor 1_protein coding [HGNC: 3166 NCBI: 1902]	pertusis	invasive bladder cancer biomarker	30867795;9804623
#N/A	ABHD14B	dirAAHB14B	chr3:51968816 -51969034	−	exonic	219	219	Abhydrolase Domain Containing 14B _protein coding [HGNC: 28235 NCBI 84836]	none	none	
hsa_drc_0128 684	HNRNPA B	circHNRNPAB	chr5:17821063 5-178210877	+	exonic	243	243	Heterogeneous Nuclear Ribonudeoprotein A/B protein coding [HGNC: 5034 NCBI: 3182]	none	none	
hsa_circ_0005 015	HAS2	circHAS2	chr8:12162871 4-121629340	−	exonic	627	627	Hyaluronan Synthase 2 _protein coding [HGNC: 4819 NCBI: 3037]	breast cancer	diabeties retinopathy biomarker	29288268; 22113945; 33954907
hsa_drc_0001 461	FAT1	circFAT1	chr4:18670656 3-186709845	−	exonic	3283	3283	FAT Atypical Cadherin 1_protein coding [HGNC: 3595 NCBI: 2195]	various cancers	pro-cancer cell stemness; pro-breast cancer drug resistance osteoblast differentiation	34314629; 34288822 35003269; 23354438;

**Table 2 ijms-25-00680-t002:** General information and links to data sources below.

Cell Line	Run	Spots (Millions)	Bases	Size	GC Content	Published	Access Type
HDLEC	Acc. Numb.pending	105.3	15.8 G	16.3 G	47.00%	15 November 2023	public
HDLEC	Acc. Numb.pending	107.7	16.1 G	16.7 G	48.00%	15 November 2023	public
HDLEC	Acc. Numb.pending	96	14.4 G	15 G	48.00%	15 November 2023	public
HUAEC	SRR3192390	200.7	40.5 G	26.1 G	56.80%	29 March 2016	public
HUAEC	SRR3192391	88.2	17.8 G	9.9 G	53.20%	29 March 2016	public
HUVEC	SRR1959051	42.8	8.3 G	5.3 G	51.80%	6 September 2016	public
HUVEC	SRR1959052	42.1	8.2 G	5.2 G	52.10%	6 September 2016	public
HCAEC	SRR9163311	12.8	3.4 G	1.0 G	54.00%	31 May 2019	public
HCAEC	SRR9163312	9.2	2.4 G	752.7 M	54.30%	31 May 2019	public
HCAEC	SRR9163313	9.3	2.5 G	755.0 M	52.90%	31 May 2019	public
HCAEC	SRR9163314	5.3	1.4 G	429.2 M	53.20%	31 May 2019	public
SAT-SVF	SRR9163319	13.5	3.6 G	1.1 G	53.40%	31 May 2019	public
SAT-SVF	SRR9163320	7.6	2.0 G	610.9 M	54.40%	31 May 2019	public
SAT-SVF	SRR9163321	10.2	2.7 G	824.3 M	54.70%	31 May 2019	public
SAT-SVF	SRR9163322	8.3	2.2 G	677.6 M	53.90%	31 May 2019	public
SMC_UA	SRR3192392	129.4	26.1 G	15.4 G	56.10%	29 March 2016	public
SMC_UA	SRR3192393	175.4	35.4 G	22.1 G	52.60%	29 March 2016	public

**Table 3 ijms-25-00680-t003:** Primer sequences.

Primer Sequences			
Cell Type Specificity	Targeted Transcripts	Forward	Reverse
All	GAPDH	TCAAGGCTGAGAACGGGAAG	CGCCCCACTTGATTTTGGAG
	circCDYL	GCTGTTAACGGGAAAGGTTGA	GTCCTCGCTGTCATAGCCTT
	CDYL	GGCTTCACCCACATCTTGTT	TACCAGCTTGCTGTCATCGG
	circZKSCAN1	AAACCCCGCCTCTTACAGTC	AAACAGGGTCTGTGCTCACC
	ZKSCAN1	ACATTCGTCTCGGAAACCCC	GGTCTCTGGGACTACCCTCA
Endothelial	circCARD6	GCAAGGAGTCCAGATGAAGACA	CCTTTCTGCTTCTATCCATGTTCA
	CARD6	CCATTTGCGGCTTAAGGCAT	TGAATTACTGCTTGCCCCCAT
	circPLXNA2	CTGTGGCCTCCTACGTTTACA	GCCCTCACATGATTCTTTTTCA
	PLXNA2	GTCAAGTGCTCCAACCCTCA	GCGATCTCGGAGAAGTCCAG
Umbilical	circFKBP8	ACAACATCAAGGCTCTCTTCCG	GTGAGCATCTCCAGGTCAGG
	FKBP8	GCCAGACAACATCAAGGCTCT	AAGGTTCCAGCTTCAGGGCT
	circERCC6L2	ACCATACAAACCAGACCACCTT	CTTCCTGAACGCCATCTGCG
	ERCC6L2	AGGGTGCATTCTGGGTGATG	CCTCACGAGTTCCCTTTTTATGC
Non-endothelial	circLPAR1	GGCTGCCATCTCTACTTCCAT	ACTCAGATAGGTGGATGGGGA
	LPAR1	GGCTGCCATCTCTACTTCCAT	AGGCAATGGACTCGTTGTAGA
HDLEC	circP2	AGACTGTGAGCTGTACAGGG	GTGCTGTCATGGTCAGGCAT
	PROX1	ATCAACGATGGGGTCACCAG	GGATCAACATCTTTGCCTGCG
	circIQSEC1	GACACAAATTACCAATACCAGGAATGAGAA	CCACACTGATATTTTTCTTCGTCCTTTGG
	IQSEC1	CTATGAGCTCTCCTCGGACCT	GTACTGGCGAAACGCCGTC
HUAEC	circFIRRE	TGCAGATACGATGCTGAGTGAA	ACTGACACCTTAGTCTCCTCA
	FIRRE	GGGAAGACTTGGTTGTGCAGAA	CCAGCCAGGATTGCTCCAGT
HUVEC	circZMYND8_A	GTCAGCTCCTATCACGACGA	ATTTATTGGAACCCAGGCCCC
	ZMYND8	TGTGAACATGAGATGAATGAAATCG	CATCGACCTGCCCGTCTTTA
	circFUT8	AGCCGAGAACTGTCCAAGAT	TATTGTCCTGTACTTCATGCGC
	FUT8	CCCACAGCCTTGGCTAGAAA	CTGTGCGTCTGACATGGACT
HCAEC	circSERPINE1	AGATCGAGGTGAACGAGAGTG	GAGTCGGGGAAGGGAGTCTT
	SERPINE1	TCGCAAGGCACCTCTGAGAAC	GCAGACCCTTCACCAAAGACA
SAT-SVF	circABHD14B	CCTCAAGCGAAGGGTCATATTTGGA	ATGAGCCTCCACACAAGCACT
	ABHD14B	GAACCTGGGTACACTGCACA	GCTGCTGCTTCCTTGGAGT
	circHNRNPAB	TTAGGCAGCGTGTGGTGTCT	CCAAACAAAGCATGTGTGCGATC
	HNRNPAB	AACCCGTGAAGAAGGTTCTGG	ATAGACTTCTTTGGGCTGGGC
SMC-UA	circHAS2	CTTCAGAGCACTGGGACGAA	TCCAAGGAGGAGAGAGACTCC
	HAS2	TGTACACAGCCTTCAGAGCA	GGCTGGGTCAAGCATAGTGT
	circFAT1	TGGTAATGACGGTGTCGGCT	GGCTGCCATCACTGTCTCCAA
	FAT1	AACCCTTGCCAGAATGGAGG	AGGAACACGGATTGACGCTT

## Data Availability

Data contained within the article.

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
