# Peer review of "Specific Circular RNA Signature of Endothelial Cells: Potential Implications in Vascular Pathophysiology"

_ijms, 2024, doi:10.3390/ijms25010680_

Round 1

Reviewer 1 Report

Comments and Suggestions for Authors

Circular RNAs (circRNAs) have been shown to exhibit tissue-specific expression profiles. In the manuscript Diallo et al. describe a study exploring circRNA expression patterns in endothelial cells from different vascular beds. The authors showed a specific circRNA signature for each endothelial cell type investigated. The work is interesting.

Specific comments

1)      According to the information presented in the table inserted between lines 159-160, all sequencing data were either recently published or retrieved from the NCBI Sequence Read Archive. Therefore, the subsection “2.1. RNA sequencing” should be removed.

2)      The data source for HDLECs was privately accessed, but it was already published as shown in the table inserted between lines 159-160. This is confusing. If the data was published, the authors should reference the publication.

3)      The manuscript has 3 tables that should be numbered and cited in the order they appear in the text.

4)      A subsection describing statistical analysis should be added to the “Materials and methods”.

5)      English editing is required to correct mistakes, e.g., a sentence that reads as “The closest addressed circular RNAs diversity in hematopoietic cells only [23] or in single ECs” (lines 71-72) lacks a predicate, “by products” (line 83) should be changed to “by-products”, “availed” (line 148) to “available”, etc.

Comments on the Quality of English Language

English editing is required to correct mistakes.

Author Response

Reviewer 1

Circular RNAs (circRNAs) have been shown to exhibit tissue-specific expression profiles. In the manuscript Diallo et al. describe a study exploring circRNA expression patterns in endothelial cells from different vascular beds. The authors showed a specific circRNA signature for each endothelial cell type investigated. The work is interesting.

Specific comments

  • According to the information presented in the table inserted between lines 159-160, all sequencing data were either recently published or retrieved from the NCBI Sequence Read Archive. Therefore, the subsection “2.1. RNA sequencing” should be removed.

This is a misunderstanding, as the HDLEC sequencing was indeed performed by us for the publication of this article. We therefore prefer to leave this section in the article.

  • The data source for HDLECs was privately accessed, but it was already published as shown in the table inserted between lines 159-160. This is confusing. If the data was published, the authors should reference the publication.

The answer to this point refers to point number 1. RNA sequencing of HDLECs was carried out by us to generate the data published in this article. The data have been submitted to the public databases (https://www.ebi.ac.uk/fg/annotare/). We are currently awaiting receipt of the access numbers (status: in progress). As soon as the access numbers are received, they will be inserted in the table to facilitate access to the data for readers of the article.

  • The manuscript has 3 tables that should be numbered and cited in the order they appear in the text.

This problem has been corrected. The figures and tables have been numbered and are cited in the order in which they appear.

  • A subsection describing statistical analysis should be added to the “Materials and methods”.

This sub-section has been added to the "Materials and methods" section.

  • English editing is required to correct mistakes, e.g., a sentence that reads as “The closest addressed circular RNAs diversity in hematopoietic cells only [23] or in single ECs” (lines 71-72) lacks a predicate, “by products” (line 83) should be changed to “by-products”, “availed” (line 148) to “available”, etc.

A thorough proofreading process was used to correct any mistakes, and we are confident that all corrections have been made.

Reviewer 2 Report

Comments and Suggestions for Authors

This study was designed to analyze RNA sequencing data from various endothelial cells to evaluate their circRNA expression profiles. The aim was to acknowledge the diversity among these cells, identify novel gene signatures, and discover endothelial cell-specific markers.

The reviewer acknowledges the detailed examination of circRNAs in this paper and recognizes its potential clinical significance. However, there are several aspects that warrant critique. There are several comments as described below. 

Major comments:

1.       The manuscript's readability could be significantly improved. The authors are advised to reorganize the content throughout the document for better clarity and flow.

2.       The integration of discussion points within the Results section hampers readability. This should be rectified by distinctly separating these sections. 

3.       The order of supplementary figures should correspond with their mention in the text for ease of reference. 

4.       While the study acknowledges the variation in circRNA secretion by different endothelial cells, its clinical relevance remains unclear. The authors should elaborate on the practical applications of these findings in clinical settings.  

5.       The conclusion of the study is not clearly articulated. The authors should explicitly state their conclusion to enhance the paper's impact. 

6.       Given that circRNAs appear to circulate systemically, the differential secretion by various endothelial cells might have a subtle yet widespread effect on the body. How do the authors address this potential impact in their study?

Author Response

Reviewer 2

This study was designed to analyze RNA sequencing data from various endothelial cells to evaluate their circRNA expression profiles. The aim was to acknowledge the diversity among these cells, identify novel gene signatures, and discover endothelial cell-specific markers.

The reviewer acknowledges the detailed examination of circRNAs in this paper and recognizes its potential clinical significance. However, there are several aspects that warrant critique. There are several comments as described below. 

Major comments:

  1. The manuscript's readability could be significantly improved. The authors are advised to reorganize the content throughout the document for better clarity and flow.

The reading of the manuscript has been improved by making the results and discussion more fluid. A thorough reorganization of the manuscript has been carried out. We hope this will satisfy the reviewer.

  1. The integration of discussion points within the Results section hampers readability. This should be rectified by distinctly separating these sections. 

Part of the results have been moved to the discussion section. The discussion section has been extensively reorganized and reordered to make it easier to read. We hope this will satisfy the reviewer.

  1. The order of supplementary figures should correspond with their mention in the text for ease of reference. 

Thank you for this point. A version error when submitting the additional figures led to this issue. The supplementary figures have been renumbered and appear in the order in which they were cited in the article.

  1. While the study acknowledges the variation in circRNA secretion by different endothelial cells, its clinical relevance remains unclear. The authors should elaborate on the practical applications of these findings in clinical settings. 

This point has been developed in more detail in the discussion, and possible biomedical applications of circular RNAs excreted into the circulation by endothelial cells have been suggested. 

  1. The conclusion of the study is not clearly articulated. The authors should explicitly state their conclusion to enhance the paper's impact. 

The conclusion has been rewritten for greater clarity and impact.

  1. Given that circRNAs appear to circulate systemically, the differential secretion by various endothelial cells might have a subtle yet widespread effect on the body. How do the authors address this potential impact in their study?

The issue raised by the reviewer is very interesting, but nonetheless complex to tackle. Our study identifies circular RNAs produced by endothelial cells, but we don't know which ones are excreted by endothelial cells and in what proportions. Our study is not aimed at answering this question, and we lack the means to do so. We have shown that some circular RNAs are common to all endothelial cells (endothelial signature), while others are specific to one type of endothelial cell. To access this type of information, we would need access to sequencing data obtained from human plasma/serum. Such data does exist, but the sequencing depths are too low to allow circular RNAs to be identified, and cannot be exploited. Furthermore, if we wished to identify which circular RNA is excreted from which type of endothelial cell, we would need to perform RNA sequencing on conditioned media from primary cultures. This type of data do not exist to our knowledge. Furthermore, the issue of circular RNA excretion by endothelial cells in the systemic circulation is approached speculatively in our article's discussion section as a gateway to future scientific endeavors. Specifically, we have focused on the biomedical applications and potential uses arising from the presence of circular RNAs in the systemic circulation.

Reviewer 3 Report

Comments and Suggestions for Authors

The manuscript by Leïla Halidou Diallo et al. presents a comprehensive study on circular RNAs (circRNAs) in endothelial cells (ECs) and their potential role in vascular biology and related diseases. The research focuses on analyzing RNA sequencing datasets from various types of ECs to identify common and distinct circRNA expression profiles. The manuscript introduces a novel area of study and emphasizes the need for deeper investigation into these complex associations. I have the following comments:

1.       Since the datasets are from both published and unpublished sources, the authors should include detailed information on how RNA sequencing data was processed, including balancing different libraries and addressing limitations like possible underestimation of read counts. And was batch correction been performed?

2.       A more straightforward explanation of the bioinformatics tools used in the study, outlining their strengths and limitations, would be helpful for readers interested in the methodology.

3.       The section on the role of circRNAs in vascular diseases could be expanded with more specific hypotheses about how these circRNAs contribute to various vascular pathologies.

4.       The potential of circRNAs in therapeutic applications for vascular diseases is only briefly touched upon, and a more in-depth discussion in this area would be helpful.

5.       Discussing the stability and detectability of circRNAs in body fluids would be helpful, particularly regarding their potential as biomarkers in clinical applications.

Author Response

Reviewer 3

The manuscript by Leïla Halidou Diallo et al. presents a comprehensive study on circular RNAs (circRNAs) in endothelial cells (ECs) and their potential role in vascular biology and related diseases. The research focuses on analyzing RNA sequencing datasets from various types of ECs to identify common and distinct circRNA expression profiles. The manuscript introduces a novel area of study and emphasizes the need for deeper investigation into these complex associations. I have the following comments:

  1. Since the datasets are from both published and unpublished sources, the authors should include detailed information on how RNA sequencing data was processed, including balancing different libraries and addressing limitations like possible underestimation of read counts. And was batch correction been performed?

All data published in the article are public. The data generated in our study (HDLEC RNASeq) are pending receipt of the accession number, which will be inserted in Table 1 as soon as it is received and before the article is published. All information on how to obtain the libraries is therefore public and available online with the data sets. We consider that this data would unnecessarily overload the article if inserted

Batch correction is used to correct for technical variations between samples. One way of detecting whether there is a batch effect is to perform a clustering analysis. This was carried out on the 250 circRNAs most highly expressed in the cell types tested. The accompanying figure shows that the data are clustered by cell type, indicating that there is no major batch effect and that the data obtained are usable. Using batch correction can also have unintended effects as it may unintentionally remove actual biological variation. For these reasons, we have not made any batch corrections. We are nevertheless aware that there are variations linked to the different technical approaches used for RNASeq, particularly sequencing depth, of the different cell types. However, we feel that these do not interfere with our analysis, in which we carry out de novo identification of new circular RNAs in the cell types tested, in order to identify gene signatures and not to quantify circular RNAs comparatively between the different cell types.

  1. A more straightforward explanation of the bioinformatics tools used in the study, outlining their strengths and limitations, would be helpful for readers interested in the methodology.

This is described in detail in the section on materials and methods.

  1. The section on the role of circRNAs in vascular diseases could be expanded with more specific hypotheses about how these circRNAs contribute to various vascular pathologies.

This point is indeed interesting. It has been further elaborated upon in the discussion section.

  1. The potential of circRNAs in therapeutic applications for vascular diseases is only briefly touched upon, and a more in-depth discussion in this area would be helpful.

This point has also been further developed in the discussion.

  1. Discussing the stability and detectability of circRNAs in body fluids would be helpful, particularly regarding their potential as biomarkers in clinical applications.

The use of circular RNAs as therapeutic biomarkers is likely the simplest and most relevant application to implement quickly. We have written a specific paragraph on this in the discussion.

Round 2

Reviewer 2 Report

Comments and Suggestions for Authors

This reviewer has no further comment. 

Comments on the Quality of English Language

N/A